# OpenReview forum: "ZeroSecBench: Fine-grained and Robust Evaluation for Secure Code Generation"
_ICLR.cc/2026/Conference — Submitted to ICLR 2026_

### Official Review · Reviewer_M6VC · 2025-10-21

**Soundness:** 1
**Presentation:** 1
**Contribution:** 1
**Rating:** 2
**Confidence:** 4

**Summary:**

This paper introduces a code security benchmark for few-line secure code generation in Java. The key innovation over prior benchmarks lie in i) using not only a CWE-level categorization of vulnerabilities, but also their assignment to and balancing over software components; and ii) introducing perturbations in the test cases to evaluate code security in noisy contexts. The data is collected from open-source repositories on GitHub.

**Strengths:**

- Important problem.
- Balancing the collected CWEs over software components lends itself for more nuanced insights over the typical security errors current coding models make.
- Evaluating code security under perturbations to natural contexts is an interesting new angle.

**Weaknesses:**

- The benchmark is limited to only one programming language and few-line completions. Such benchmarks have already been overtaken in the current literature, and as such, this benchmark seems to lack sufficient novelty. The component-wise categorization is unclear, insufficient in itself for a strong contribution, under-discussed and under-explored in this paper; and most crucially, unclear if a similar analysis would not have been otherwise possible on existing benchmarks.
- The data collection and construction pipeline is described only vaguely in the paper, with several steps remaining unclear, e.g., how vulnerabilities are identified and verified, how functionality test cases are built, and how exactly the perturbations of stage 4 are constructed. Currently, it reads as if the vulnerabilities used for the benchmark are freshly identified from the examined repositories. This would imply that (I hope) that the authors have informed the maintainers of these repositories about the security issues found (however, I see no mention of this in the paper, which is a serious concern of mine).
- The benchmark quality evaluation relies on non-standard, often non-sensible metrics over-rewarding the current submission (e.g., tasks per programming language favoring this benchmark as it only consists of a single programming language).
- The evaluation is lacking in depth and quality. In terms of depth, the key issue is the lack of a systematic analysis of model performance across varying perturbations for a given test case---this could have been otherwise one of the key novel findings of the paper. In terms of quality; the conclusions under Table 3 state that GPT-5, DS R1, and Gemini 2.5 Pro are all better than than GPT-5-mini, DeepSeek V3 , and Gemini 2.5 Flash, but the stated metrics are often higher for these models contradicting the statement. In general, the evaluation looks dubious, as most models score extremely closely to eachother. Further, in 4.2 it is stated that instruct and autocomplete scenario score similarly, but then in Figure 4, there are visibly large differences. Finally, while per-component scores are provided, there is no actionable takeaway or any clear insight to derive apart from the model varying and evaluation should incorporate this. Why so? Is the variance maybe also related to the components?
- The paper uses a wrong font, not the template.

**Questions:**

Based on what is the claim that BaxBench requires on average 7.1 lines per task to complete made? The benchmark requires the generation of complete webapps with several endpoints, this averaging to only 7.1 LoC sounds highly implausible.

**Details Of Ethics Concerns:**

At the moment, I am not entirely sure about the dataset construction, so I am not raising an ethical concern. But if my understanding is correct that the benchmark has found new vulnerabilities in open-source without responsibly disclosing them to the authors, I am concerned.

---

### Official Review · Reviewer_fmgU · 2025-10-30

**Soundness:** 2
**Presentation:** 3
**Contribution:** 2
**Rating:** 2
**Confidence:** 4

**Summary:**

This paper introduces ZeroSecBench, a benchmark designed for fine-grained and robust evaluation of secure code generation in AI copilots. Unlike prior benchmarks that rely solely on CWE-level granularity and simplistic datasets, ZeroSecBench adds a CWE × component × scenario taxonomy and employs five robustness-oriented augmentations. The benchmark consists of 850 vulnerability instances mined from 150,000 real-world GitHub repositories, covering 12 CWEs and 46 Java components, with both autocomplete and instruct settings. Evaluation results show low overall secure pass@1, and significant disparities across vulnerability types.

**Strengths:**

1. Introduces a three-axis taxonomy that combines CWE, affected component, and vulnerability scenario, enabling more fine-grained evaluation.
2. The five augmentation strategies significantly improve the benchmark’s diversity and robustness compared to existing datasets.

**Weaknesses:**

1. The paper does not demonstrate strong technical novelty.
2. The overall design and methodology are not clearly presented.

**Questions:**

This paper introduces an interesting benchmark. However, in its current form, the contributions may not be sufficient for acceptance.
1. The work does not strongly highlight technical novelty, and it would help if the key innovative aspects were emphasized more clearly.
2. For a security benchmark, CWE coverage is a fundamental baseline. The addition of affected components and vulnerability scenarios is promising, but it would strengthen the paper to show how these additions improve benchmark quality. An ablation study could also illustrate the value of the five robustness strategies and whether all are necessary.
3. The papers choses to build a new dataset from scratch. A discussion of why this approach was taken, instead of extending existing benchmarks (such as SecCodePLT), would provide useful context.
4. The involvement of security experts is a strong aspect of the work, as they review test cases and design rules for the AST scanner. Providing more details on how these validations or designs were conducted and how correctness was ensured would make the contribution more convincing. In addition, it would be helpful to discuss the potential challenges of extending the benchmark to other languages (e.g., Python or C), since such an expansion may require substantial manual effort.
5. Some design aspects remain unclear, such as which AST scanner was used, how the dynamic execution environment was implemented, how natural-language instructions were created, and how data leakage was prevented, etc. Clarifying these points would improve reproducibility and confidence in the dataset.

---

### Official Review · Reviewer_1buw · 2025-11-01

**Soundness:** 2
**Presentation:** 2
**Contribution:** 3
**Rating:** 4
**Confidence:** 4

**Summary:**

The paper introduces ZeroSecBench, a new benchmark for evaluating the security of AI code-generation models. ZeroSecBench proposes a fine-grained three-axis labeling scheme — mapping each vulnerable instance to a CWE category, affected software component (library/framework/API), and concrete vulnerability scenario. The dataset is constructed from large-scale real-world Java repositories, leveraging AST-based vulnerability mining with expert-defined rules, followed by multi-stage robustness enhancement. The benchmark supports both Autocomplete and Instruction-following workflows, and introduces a three-stage hybrid evaluation pipeline (syntax → functionality → security), where the last stage includes LLM-as-Judge voting and dynamic PoC execution. For the SOTA model they evaluate, the resolved rate might still be low(26%).

**Strengths:**

1. Promising direction. Establishing the benchmark to highlight the security risks associated with Code GenAI is a direction worth studying.

2. Collect real-world security problems from the OSS repo to make the task more realistic.

3. Robustness-enhanced design is interesting and creative.

**Weaknesses:**

1. For the data collection, it is unclear how they utilize static analysis (AST + rules) to do data filtering from the repo and how to handle false positives. Given that this step determines the correctness of the entire dataset, the absence of technical details would cause confusion.

2. Language and domain limitation. The benchmark is currently restricted to Java and 12 CWE categories. Results might be limited to only one language. Cause the same CWE in a different language might have different characters for LLM to understand and resolve, especially since the paper is highlighting multiple axes.

3. Dynamic evaluation coverage is narrow. Only 54 dynamic cases (17 components, 9 CWEs) are included, which limits the strength of conclusions about runtime security behavior. I appreciate that you already consider majority voting in LLM-as-Judge, but it is still not always reliable and verifiable. Besides, we still need to have a better ablation study about how LLMs perform consistency in LLM-as-Judge.

**Questions:**

As mentioned in the weakness.

---

### Official Review · Reviewer_5HB4 · 2025-11-05

**Soundness:** 2
**Presentation:** 2
**Contribution:** 2
**Rating:** 2
**Confidence:** 4

**Summary:**

This paper introduces ZeroSecBench, a benchmark designed for fine-grained and robust evaluation of secure code generation in LLM-based AI copilots. The authors argue that existing benchmarks are limited by coarse-grained CWE-only labeling and insufficient robustness. To address this, they propose a three-axis taxonomy (CWE → component → scenario) and apply five robustness augmentations (mask-position variation, unsafe-code distractors, grammatical traps, contextual noise, and data leakage prevention). ZeroSecBench includes 850 instances from 150k GitHub repositories across 12 CWEs and 46 Java components, evaluated under both autocomplete and instruct settings. The paper reports evaluation results for 11 modern LLMs (e.g., GPT-5, Deepseek-R1, Gemini-2.5, Claude 4) and shows low overall security pass rates (>0.26).

**Strengths:**

1. Timely Topic: Secure code generation is an increasingly important problem as AI copilots are widely adopted. The paper’s focus on this issue is relevant to both academia and industry.


2. Motivation Clarity: The paper clearly motivates the need for more granular and robust benchmarks beyond CWE-level classification.


3. Breadth of Comparison: The benchmark is compared to 13 prior works along multiple design dimensions, providing a broad context.


4. Comprehensive Evaluation: Testing 11 state-of-the-art models offers a large-scale empirical picture of current LLM security weaknesses.

**Weaknesses:**

1. Single-Domain Scope: Despite the claim of generality, the benchmark only covers Java and 12 CWEs. This narrow scope substantially limits its impact and validity for “robust” or “cross-domain” evaluation. The contribution of considering “component” seems also Java-related.


2. Insufficient Validation of Robustness Claims: Although five augmentation strategies are proposed, their effectiveness is not empirically verified (e.g., ablation or sensitivity analysis). It’s unclear whether these augmentations genuinely make evaluation more realistic or simply unnatural.


3. Reliance on LLM-as-Judge: Security evaluation partly depends on another LLM’s voting, which may introduce bias and temporal inconsistency. No human validation or reliability check is provided to justify this choice.


4. Limited Insight from Results: The results largely restate that “security performance is low and varies by component”. This finding, while true, is not analytically deep or novel.


5. Overclaiming Benchmark Quality – The “highest quality score” comparison across 13 benchmarks is not well-founded; the scoring system is self-defined and lacks independent validation.


6. The performance difference between the evaluated models is very small. Therefore, it is unclear whether ZeroSecBench is suitable for measuring model progress. In the worst case, it could happen that the model includes two extreme cases: easy samples that every model could solve and hard samples that no model could solve. It would be helpful if the paper could provide more insight how solvable and diverse ZeroSecBench is.

**Questions:**

Please address my comments in the "Weaknesses" section.

---

### Meta-Review · Area_Chair_M6va · 2026-01-07

**Summary:**

Reviewers have concerns about the limited scope and questionable quality of the proposed benchmark dataset, insufficient justification regarding its enhanced robustness, lack of validation of its evaluation, lack of in-depth analysis, and the unclear implications of the dataset.

**Reviewer Concerns:**

No rebuttal response.

**Reviewer Scores:**

No rebuttal response.

---

### Decision · Program_Chairs · 2026-01-26

Reject